# Geographic patterns of upward shifts in treeline vegetation across western North America, 1984–2017

Joanna L. Corimanya<sup>1</sup>, Daniel Jiménez-García<sup>2,3</sup>, Xingong Li<sup>4</sup>, and A. Townsend Peterson<sup>1</sup>

**Correspondence:** Joanna L. Corimanya (gresham.joanna@gmail.com)

- **Abstract.** Previous research has shown that (1) treelines are shifting upward in elevation on high mountain peaks worldwide,
- and (2) the rate of the upward shift appears to have increased markedly in recent decades, at least in a few cases that have been
- studied in detail. Because treeline elevational shift is a process manifested over broad scales of space and time, a particular
- challenge has been that of obtaining a broad-enough view of patterns of treeline shift to permit inferences about geographic
- and environmental patterns. What is more, intensive studies of treelines have been concentrated in North Temperate regions,
- such that little information is available about treeline shift patterns at lower latitudes. We attempted to address this challenge by
- analyzing long time series of vegetation indices derived from Landsat imagery obtained and prepared via Google Earth Engine 7
- from the 1980s to the present. We sampled vegetation indices at points spaced every 100 m along 100 km transects radiating
- out in eight directions from 115 high peaks across western North America (Canada to Central America), which means that we
- 10
- are sampling approximately every second or third pixel in the corresponding Landsat images. Considerable data preparation

was necessary, including ending transects 

<sup>&</sup>lt;sup>2</sup>Laboratorio de Biodiversidad, Centro de Agroecología y Ambiente, Instituto de Ciencias, Benemérita Universidad Autónoma de Puebla, Puebla, Mexico

<sup>&</sup>lt;sup>3</sup>Laboratorio Nacional CONAHCYT de Biología del Cambio Climático, Mexico

<sup>&</sup>lt;sup>4</sup>Department of Geography, University of Kansas, Lawrence, Kansas, USA

# 19 1 Introduction

The upper elevational limits of forests in mountain systems represent a fascinating and dramatic manifestation of distributional limitation at the species and community levels. Treeline phenomena have seen extensive analysis and discussion in the eco-logical literature: they are an important manifestation of the geographic ecology of ecosystems, and likely reflect important climate-related controls (Kullman, 1998). Numerous studies have been developed that aim to understand factors driving the location and possible shifts in treelines, with the general conclusion that treelines are determined by complex suites of fac-tors (Cudlín et al., 2017; Körner, 1998; Holtmeier and Broll, 2005; Irl et al., 2016; Grafius et al., 2012; Kienle et al., 2023). Whereas some researchers have concluded that treeline position can be distilled down to simple rules regarding seasonal mean ground temperatures (Körner and Paulsen, 2004), others have argued that treeline drivers are considerably more multidimensional and complex (Paulsen and Körner, 2014; Zhao et al., 2015). In this study, we adopt Körner's (2012) definition of elevational treeline, i.e., the uppermost elevation on a mountain slope at which upright woody plants (trees >2m tall) can maintain self-sustaining populations. Above that limit, insufficient warmth (a too short or too cold growing season) prohibits the regular recruitment and survival of true tree forms, even if isolated, krummholz-like individuals occur sporadically.

Clearly, considerable complexity is involved in any attempt to characterize treeline phenomena. However, dendroecological approaches offer the useful possibility of obtaining establishment ages on an individual-tree basis across broad stands of trees at or near treelines (Elliott, 2011). When treelines change, a key challenge is that of considering treeline shifts (i.e., elevational advance upward with warming climate) versus densification (i.e., sparse forest or scattered trees near treeline filling in with more trees, regardless of whether the treeline changes or not) (Shi et al., 2022). Finally, treeline is a highly scale-dependent phenomenon, such that all of its qualities vary in importance and effect at different spatial extents and resolutions (Holtmeier and Broll, 2017).

From early in discussions about the possibility that global climates would warm with increasing greenhouse gas concentrations (LaMarche et al., 1984; Grace et al., 2002), the expectation has been that treelines would advance up mountain slopes as climatic controls relax at extreme elevations. Empirical evidence has been mixed, however, with some studies documenting what appears to be very rapid treeline advance (Peterson et al., 2022), and others finding no evidence of overall tendency to change (Beloiu et al., 2022). One broad analysis found that treeline advance was faster in subarctic regions than in temperate regions (Lu et al., 2021), and another found that treelines experiencing stronger winter warming and with diffuse treeline forms were more likely to advance (He et al., 2023).

Nonetheless, most of these previous broad-scale analyses of patterns of treeline advance in the face of warming climates have been based on datasets with strong inherent biases and significant gaps. That is, in largest part, treeline studies have been conducted in the North Temperate zone: examples of such biased analyses are many (Shi et al., 2022; Zhao et al., 2015; Körner, 1998; Lu et al., 2021). A few analyses have achieved a somewhat better balance of representation of treelines in the Tropics and in the Southern Hemisphere (He et al., 2023; Hansson et al., 2023; Kienle et al., 2023). The concern, of course, is that such information gaps and biases in what information is available may blind researchers and their analyses to very real and important patterns in the global occurrence of the phenomenon of treeline advance.

Here, we address these important knowledge gaps about treeline dynamics in the face of warming climates globally over the past several decades. We assess the null hypothesis that magnitude of alpine treeline shifts is not related to a series of geographic features, such as latitude, longitude, and distance to coast. Specifically, to be able to assess treeline shifts on a continent-wide basis, we use a long time series of remote-sensing data to seek patterns in the magnitude of alpine treeline shifts across 115 high peaks in western North America, from Central America to southern Canada. We use vegetation index trends along transects radiating out from each peak in eight cardinal and sub-cardinal directions; the vegetation index approach has the advantage of "seeing" vegetative mass generally, in effect integrating over both treeline advance and densification of sparse, near-treeline forests (Feuillet et al., 2020). Of course, these broad-scale analyses are not a substitute for more detailed, field-based analyses, nor should vegetation-index-based assessments replace more fine-grained inspections of the actual geometry of treelines. The result is a novel dataset from which we have derived several intriguing insights about geographic patterns in the magnitude of treeline elevational shifts.

# 2 Methods

# 2.1 Mountain peak characterization

Our aim was to characterize temporal changes in vegetation mass on a set of mountains that covered western North America. To that end, we chose to follow a comprehensive summary of high mountains worldwide (Maizlish, 2007), which is based on an effort to identify all mountains worldwide with at least a 1500 m prominence; the authors of that compendium (called the Ultras Project) researched all summits on Earth that meet this criterion, finding 1524 such peaks. From that worldwide dataset, we extracted the 354 mountain peaks located in North America (Panama to Canada). We used the coordinates of each peak in the peaks dataset as a centerpoint, and plotted eight transects in each of the cardinal and sub-cardinal directions extending out from that centerpoint; points were plotted and distances measured in meters using the WGS84 Special Mercator for Web Applications (EPSG:3857) projection to assure consistent distances among sampling stations. Transects were each initially 100km long, with sampling stations every 100m, so each transect included 1000 sampling stations.

We excluded from analysis all mountains that were forested to the peak, or that showed signs of anthropogenic modification at or around the peak upon visual inspection of the region in Google Earth. We also excluded peaks for which treelines were not associated clearly with the upper slopes of the peak, but rather were lower, extending just a bit up the valley walls and thus likely represent latitudinal treelines as opposed to altitudinal treelines; such low treelines were particularly common in central and northern Canada and Alaska, such that all of the far northern peaks were excluded. Given that, in eastern North America, only one peak (Mt. Washington, in New Hampshire) met our criteria, to avoid including a genuine spatial outlier in our analyses, we omitted that peak from analysis, thus focusing our analyses on the high peaks of western North America. At the end of this process, from 354 peaks in the initial database, we had 119 peaks remaining as a basis for our analyses (**Figure 1**).

**Figure 1.** The 119 high mountain peaks analyzed in this study. Triangles represent individual mountain peaks used in our analysis. The 'X' symbol is Mt. Washington, which was removed from the dataset prior to analysis. This map was constructed using QGIS ver 3.38.2. The ESRI physical basemap was used to create the map.

In © Google Earth Engine, we overlaid the transect sampling points on imagery from Landsat for the period 1984–2017, and 84 associated the values of the normalized difference vegetation index (NDVI) with each sampling point in the transect dataset. 85 86 For this analysis, we focused on early (1984–1988) and late (2013–2017) time periods within the timespan of the Landsat dataset to create a before-and-after contrast. We used NDVI data from the annual Landsat collection (Landsat/LT5\\_L1T\ 87 ANNUAL\ NDVI, Landsat/LE7\ L1T\ ANNUAL\ NDVI, and Landsat/LE8\ L1T\ ANNUAL\ NDVI) in © © 88 89 Google Earth Engine. We used the pre-processed LANDSAT\ LT5\ L1T\ ANNUAL\ NDVI collection, which provides annual NDVI composites derived from Level-1 terrain-corrected Landsat 5 reflectance images (including cloud masking and 90 91 quality assurance; unfortunately, this collection is now deprecated in GEE). To ensure full transparency, our scripts for reproducing the NDVI computation from the original Landsat reflectance data are publicly available on Github (see below). Detailed 92 93 information about the original dataset can be found in the Earth Engine Data Catalogue (USGS, 2017); calibration procedures and validation methods for this collection are described by Chander et al. 2009. We generated a composite for each year from 94 95 the available Landsat images, and extracted NDVI values for each year via a mean reducer. We then inspected each transect of 96 each peak individually by overlaying the point data on the Google Satellite fine-resolution data product, using the GIS capabilities of QGIS (version 3.2). Similarly, we extracted elevation at each sampling station within each transect using the NASA 97 Shuttle Radar Topography Mission 30m resolution Digital Elevation Model in © Google Earth Engine (Farr et al., 2007). 98

A key step was that of choosing the sampling station along each transect that corresponded to treeline, as follows. Descending from each peak along each transect, via reference to the © Google Satellite data layer in QGIS, we identified the sampling station that most closely approximated the upper elevational limit of forest. That is, we ignored single, isolated trees, but rather identified the elevation at which forest became continuous, albeit in some cases still sparse. For this sampling station, we set the field TreesBegin in the data table characterizing peaks to 1.

# 2.2 Data refinement

99

101

102

103

104

115116

117

Values of NDVI and elevation were assigned to each sampling station via GIS overlay ("extract value to point") operations.

All subsequent data preparation was done in R (version 4.4.1) and QGIS (version 3.38.2). We cleaned the data that had been

exported from Google Earth Engine by removing any missing values. We averaged the yearly NDVI values over the two time

periods (1984–1988 and 2013–2017) to provide "before and after" comparisons that would be more immune to measurement

error or other sources of year-to-year variation.

Our next goal was to calculate regression equations for individual mountains, slopes, and time periods, characterizing the negative-sloped relationship between elevation and NDVI. To this end, we transformed the data into a hierarchical nested list of lists; the dataset included 120 mountain peaks, each of which had 1-8 transects. Each transect had the two averaged year groups of NDVI data, for a total of 1848 distinct combinations of peak, transect, and year group; some transects were removed entirely based on the criteria listed above (**Section 2.1**). In our analyses, we included only NDVI measurements from stations that were in relatively close proximity to treeline. That is, we included at least the last 10 stations. If twice the number of stations after the manually identified treeline to the transect edge (i.e., the furthest measured station downslope) plus one (to explicitly account for the station representing treeline itself) exceeded 10, we used this greater number of stations instead.

When the latter calculation was greater than 10, this resulted in an equal number of points above and below treeline. This approach ensured that we captured sufficient data from both sides of the treeline, and minimized the effect of terrain variability from sources such as small bare peaks, increasing the probability of detecting true relationships.

We modeled the NDVI-elevation relationships with NDVI as the response variable and elevation as the predictor variable to find the best type of regression equation and, ultimately, the best approximation to the true relationship between these variables (**Figure 2**; see below). These models allowed us to associate NDVI and treeline elevation for calculation of our final response variables: change in elevation and change in NDVI. We calculated three types of regressions on each data frame (linear, reciprocal-linear, and reciprocal-quadratic) to assess which model shape best describes the NDVI-elevation relationship. The three models were compared via the Akaike Information Criterion (AIC; Akaike 2011) for each peak, transect, and time period. As all 1848 of these NDVI elevation relationships were best described by a linear model, we retained only linear regression equations for subsequent analyses. We excluded transects for which the regression equation was not statistically significant or for which the regression slope was positive; we used  $\alpha = 0.05$  as the threshold for statistical significance in all regressions. These latter criteria removed 688 of 1848 transects, leaving 1160 transects for analysis. Finally, since our goal was to create temporal comparisons between the two time periods, we also removed any transects for which regressions for either time period did not meet the criteria outlined above; this filter removed another 202 transects from analysis. The final dataset thus included 958 transects on 115 peaks.

**Figure 2.** Map showing continentwide patterns of regression slopes relating NDVI to elevation for each peak, averaged across the 1-8 transects available for each peak, for the 2013–2017 time period. Yellow circles represent a positive slope (excluded from final analysis), and blue circles represent a negative slope. The size of the circles coincide with the magnitude of the absolute value of each slope calculation. The 'X' symbol is Mt. Washington, which was removed from the dataset prior to analysis. This map was constructed using QGIS ver 3.38.2. ESRI physical basemap was used to create the map.

The goal in these analyses was to calculate change in treeline elevation for use as a response variable in continent-wide models. Alpine treeline position represents a bioclimatic threshold: trees cannot form self-sustaining closed canopy stands above it because low temperatures and a short growing season limit carbon assimilation and wood formation (Körner, 2012; Holtmeier, 2009). In turn, shifts in treeline elevation over time serve as a direct indicator of how local thermal regimes and associated growing-season lengths are changing on the landscape (Körner, 2012; Holtmeier and Broll, 2005). By modeling change in treeline elevation, we capture how climate warming and other environmental drivers are pushing the arboreal "limit" upslope. To this end, we inserted the elevations at our manually selected treeline position into the 2013–2017 NDVI linear regression equations to calculate the NDVI values associated with present day treeline. We then inserted that calculated NDVI value into the 1984–1988 regression equations to obtain an estimate of treeline elevation (i.e., we sought the elevation with the same 1984–1988 NDVI value as present-day treeline on that slope of that mountain; **Figure 3**). Finally, we subtracted 1984–1988 treeline elevation values from the 2013–2017 treeline elevation values to estimate the change in treeline elevation over the broad temporal span of this study.

**Figure 3.** Example of a high mountain (Cerro de la Malinche, Tlaxcala, Mexico) and inferences deriving from it regarding position of treeline through time. Top panel: View of the mountain in Google Earth, with eight transects radiating out from the peak in cardinal and subcardinal directions. White dots indicate stations at which NDVI values were sampled; purple stars indicate the position of treeline identified visually. Bottom panel: dark red points and lines show the NDVI-elevation relationship in the 1980s; blue points and lines show the same relationship in the 2010s. In one example (northward transect), the elevation of treeline observed for 2013–2017 (3960 m) was used to identify a treeline NDVI threshold (0.3135), which was in turn used to identify a likely elevation (3448 m) of the same NDVI level for 1980s conditions. Background of top panel is from Google Earth©.

We also calculated a second, simpler response variable, which was simply change in NDVI at the 2013–2017 treeline. In high-elevation contexts, an upward trend in NDVI within elevation bands at the present day treeline signals increased tree recruitment, shrub encroachment, or earlier green-up (Harsch et al., 2009; Rupp and Starfield, 2001). Thus, by computing the change in NDVI over our study period, we capture a functional or "greenness" dimension of treeline dynamics that complements the structural dimension (change in elevation). In other words, even before trees form closed canopies, small shrubs or seedlings may begin to photosynthesize more vigorously, which will manifeset as an increase in NDVI. To this end, we inserted the manually located 2017 treeline elevation into the two regression equations for that mountain and slope. This resulted in NDVI values at a particular elevation (i.e., recent treeline) for 2013–2017 and 1984–1988 for each peak and direction. We subtracted the 1984–1988 values from the 2013–2017 values to obtain the change in treeline NDVI. A more positive value for change in NDVI indicates an increase in NDVI between 2013–2017 and 1984–1988.

Finally, we assembled a suite of independent variables that may be of interest as possible drivers of variation in rates of treeline shift. We included (1) the number of stations in the transect below treeline (as a potential confounding factor), (2) cardinal direction of the transect, (3) latitude, and (4) longitude, all of which were derived from the original data about each transect and peak in the analysis. We also calculated (5) the distance to the closest coastline (in meters), based on the coastline corresponding to official maritime boundaries (Flanders Marine Institute, 2012). We built a raster file that contained the distance to the closest coastline for each pixel (1.53km resolution). We added these distance values to the data table for the transect sampling points using the point sampling tool in QGIS.

### 2.2.1 Model selection

To understand which of the above independent variables likely drive(s) variation in rate of treeline elevational shifts, we used an iterative stepwise model selection process. We selected the model that best describes western North American geographic treeline elevational shift patterns using AIC. We explored two statistical models to ensure that the final model would best explain geographic variation in treeline dynamics. First, we built 18 linear mixed models, each of which contained a random effect of 'Peak ID' to account for variability in local landscape characteristics. Second, we constructed 18 spatial mixed models using the R package 'spaMM' in which we specified Matèrn random effects to account for spatial autocorrelation by capturing spatially structured variation in treeline elevation that is not explained by the fixed effects (Rousset and Ferdy, 2014). All of these models were fitted using restricted maximum likelihood.

For the first two model sets (total 32 models), the response variable was the change in treeline elevation between the two time periods. We produced a second array of models, parallel to the first, in which we used change in treeline NDVI as the response variable. All other model characteristics were the same as for the models based on change in treeline elevation.

For all of the models described above, the fixed effects were different combinations of the independent variables: distance to coast, number of stations after treeline, cardinal direction of slope, latitude, and longitude, as well as the interaction between latitude and longitude. The models ranged in complexity, but we constrained the analysis to always include latitude and longitude. We compared all 32 models in an AIC table, as the response variable was constant and all models were fit by REML. We assessed significance by checking whether or not the 95% confidence interval of each fixed effect overlapped zero (Browne,

**Figure 4.** Panel (a) diagrams the steps taken to (1) characterize mountains, (2) clean the data in preparation for analysis, and (3) select models. In panel (b), the hierarchical structure of our dataset is conceptually illustrated.

1979). We considered results for which confidence intervals did not overlap zero to be significant. Our dataset construction and analysis steps are summarized in a diagram for clarity (**Figure 4**).

# 182 3 Results

### 3.1 Generalities about Treelines

Treeline locations were non-random in a number of ways. On average, across all mountain peaks in our analyses, treeline was located at 2433 m. However, treeline position varied systematically, in that a significant relationship existed between treeline and latitude: tropical treelines averaged 3177 m, whereas temperate-zone treelines were lower, at 2244 m. As such, all subsequent analyses in this study needed to be conditioned on the geographic complexity underlying the phenomenon of treeline.

# 3.2 Change in Treeline Elevation

Treelines have been changing, even over the relatively short, 30-40-year timespan of this study. Indeed, treeline shifts among the western North American peaks in this study had a mean overall shift of 20.2 m upslope. The distribution of change values ranged from 165 m downslope to 127 m upslope.

For the multivariate models relating change in treeline elevation to environmental drivers, we calculated the best-fit models for the linear mixed models and spatial mixed models using AIC and the coefficient of determination ( $R^2$ ). We calculated the Marginal and  $ConditionalR^2$  values for linear mixed models and a  $Pseudo-R^2$  value for the spatial mixed models. The best linear mixed model included number of stations after treeline, direction of transect moving away from the peak, latitude, longitude, and the interaction between latitude and longitude as fixed effects, with mountain peak name as a random intercept (Table A1). From our candidate set of spatial mixed models, the best fit included only latitude as a fixed effect, with a Matèrn random effect structure (Table A2). When comparing all models and the two best fitting models from the linear and spatial analyses, the spatial mixed model was best overall (Tables A3 & 1).

| Model Type | Terms                                                        | AIC | Delta AIC | Weight |
|------------|--------------------------------------------------------------|-----|-----------|--------|
| Spatial    | Latitude                                                     | 6.9 | 0.000     | 1.0    |
| Linear     | # Stations After Treeline + Direction + Latitude * Longitude | 6.9 | 2.0       | 3.8    |

**Table 1.** AIC table comparing the best linear mixed model and the best spatial mixed model from their respective comparisons, which had change in treeline elevation as the response variable. There were 2 models in this comparison.

The best spatial mixed model, which was also the best model overall, showed that change in treeline was not significantly related to the only fixed effect, latitude ( $Pseudo - R^2 = 0.4512$ ). This model was fit using a Gaussian random effect with a Matèrn correlation structure. The smoothness parameter ( $\nu$ ) was estimated at 0.398, indicating a moderate degree of spatial continuity in treeline elevation changes. The range parameter ( $\rho$ ) was 0.00466, suggesting that spatial correlation between observations declines sharply over very short distances. The variance of the spatial random effect ( $\lambda$ ) was estimated at 3,651,000, highlighting substantial spatial variation in the data. The residual variance ( $\phi$ ) was 64,159, representing variability unexplained after accounting for spatial effects (**Table 2**).

| Term                         | Estimate             | SE                  | Lower 95% CI         | Upper 95% CI        |
|------------------------------|----------------------|---------------------|----------------------|---------------------|
| Intercept                    | $2.6 \times 10^{3}$  | $2.0 \times 10^{3}$ | $-5.1 \times 10^{3}$ | $9.9 \times 10^{3}$ |
| Latitude                     | $-5.1 \times 10^{1}$ | $3.1 \times 10^{1}$ | $-1.3 \times 10^{2}$ | $1.0 \times 10^{1}$ |
| Random intercept (variance)  | $3.7 \times 10^{6}$  |                     |                      |                     |
| Random intercept (std. dev.) | $1.9 \times 10^{3}$  |                     |                      |                     |
| Residual (variance)          | $6.4 \times 10^4$    |                     |                      |                     |
| Residual (std. dev.)         | $2.5 \times 10^2$    |                     |                      |                     |

**Table 2.** Model summary of the top spatial mixed model. Fixed and random effect outputs are shown. The response variable for this model was the change in treeline elevation. Significance would be denoted by bold text and was assessed by observing whether or not the confidence interval overlapped zero, but this model found no significant relationships.

The less optimal best linear mixed model can be explored as well. It showed a significant relationship between change in treeline and latitude, longitude, and the interaction between latitude and longitude ( $ConditionalR^2 = 0.6887$ ,  $MarginalR^2 = 0.2815$ ). Change in treeline elevation was significantly higher at lower latitudes ( $\beta = -100.6$ , 95% CI = [-155.1, -46.29], **Table 3**; **Figure 5a**). The relationship between change in treeline elevation and the interaction between latitude and longitude was also significantly negative ( $\beta = -0.8418$ , 95% CI = [-1.345, -0.3413], **Table 3**): as longitude increases (eastward), effects of latitude on treeline shift become more negative, suggesting a complex spatial relationship between these geographic variables and treeline dynamics. Longitude alone also had a significant positive relationship with change in treeline elevation ( $\beta = 36.21$ , 95% CI = [13.00, 59.52], **3**; **Figure 5b**). This result indicates that mountain treelines further east in North America (farther from the Pacific Coast) have more drastic temporal changes in their treeline elevations compared to the more western mountain treelines in our study.

| Term                         | Estimate                       | SE                             | Lower 95% CI                   | Upper 95% CI        |
|------------------------------|--------------------------------|--------------------------------|--------------------------------|---------------------|
| Intercept                    | $4.2\times 10^3$               | $1.1\times10^3$                | $\boldsymbol{2.0\times10^3}$   | $6.4 	imes 10^3$    |
| # Stations After Treeline    | $6.3 \times 10^{-1}$           | 1.4                            | -2.1                           | 3.4                 |
| Direction (North)            | $1.9 \times 10^{1}$            | $5.3 \times 10^1$              | $-8.3 \times 10^{1}$           | $1.2 \times 10^2$   |
| Direction (Northeast)        | -5.0                           | $5.1 \times 10^{1}$            | $-1.0 \times 10^{2}$           | $9.5 \times 10^{1}$ |
| Direction (Northwest)        | $6.4 \times 10^{1}$            | $4.9 \times 10^1$              | $-3.1 \times 10^{1}$           | $1.6 \times 10^{2}$ |
| Direction (South)            | $7.5 \times 10^1$              | $4.8 \times 10^{1}$            | $-1.9 \times 10^{1}$           | $1.7 \times 10^2$   |
| Direction (Southeast)        | $3.3 \times 10^{1}$            | $5.0 \times 10^{1}$            | $-6.6 \times 10^{1}$           | $1.3 \times 10^{2}$ |
| Direction (Southwest)        | $4.7 \times 10^1$              | $5.0 \times 10^{1}$            | $-4.9 \times 10^{1}$           | $1.4 \times 10^2$   |
| Direction (West)             | $5.1 \times 10^1$              | $5.0 \times 10^{1}$            | $-4.5 \times 10^{1}$           | $1.5 \times 10^{2}$ |
| Latitude                     | $-1.0\times10^{2}$             | $\boldsymbol{2.8\times10^{1}}$ | $-1.6\times10^2$               | $-4.6\times10^{1}$  |
| Longitude                    | $\boldsymbol{3.6\times10^{1}}$ | $\boldsymbol{1.2\times10^{1}}$ | $\boldsymbol{1.3\times10^{1}}$ | $6.0\times10^{1}$   |
| Latitude × Longitude         | $-8.4\times10^{-1}$            | $2.6\times10^{-1}$             | -1.3                           | $-3.4\times10^{-1}$ |
| Random intercept (variance)  | $9.0 \times 10^{4}$            |                                |                                |                     |
| Random intercept (std. dev.) | $3.0 \times 10^2$              |                                |                                |                     |
| Residual (variance)          | $6.9 \times 10^4$              |                                |                                |                     |
| Residual (std. dev.)         | $2.6 \times 10^2$              |                                |                                |                     |

**Table 3.** Model summary of the top linear mixed model. Fixed and random effect outputs are shown. The response variable for this model was change in treeline elevation. Significance is denoted by bold text, and was assessed by observing whether or not the confidence interval overlapped zero.

**Figure 5.** Summary of univariate relationships between treeline elevational shifts and latitude and longitude. Panel (a) shows latitude on the x-axis, while panel (b) shows longitude on the x-axis. Regression lines for both panels are denoted in black. Note that the interaction term between these two independent variables is also statistically significant.

# 218 3.3 Change in Treeline NDVI

As with the previous response variable, we fit a series of linear mixed models and spatial mixed models with a Màtern random effect structure for change in treeline NDVI as a response variable, and compared the resulting models via AIC, both individually and in totality. The top linear mixed model ( $MarginalR^2 = 0.3300$ ,  $ConditionalR^2 = 0.7349$ ) and the top spatial mixed model had only latitude as predictor variables when compared only to models of their respective type (**Tables A4 & A5**). The best-fitting model when comparing all linear and spatial mixed models and when comparing the top models from the spatial mixed model and linear mixed model AIC tables was the spatial mixed model with the fixed effect of latitude ( $Pseudo - R^2 = 0.6067$ ; **Tables A6 & 4**).

| Model Type | Terms    | AIC  | Delta AIC | Weight |
|------------|----------|------|-----------|--------|
| Spatial    | Latitude | -1.5 | 0.00      | 1.0    |
| Linear     | Latitude | -1.5 | 1.5       | 5.3    |

**Table 4.** AIC table comparing the best linear mixed model and the best spatial mixed model from their respective comparisons, which had change in treeline NDVI as the response variable. Two models were featured in this comparison.

The best-fit linear mixed model revealed that change in treeline NDVI was significantly related only to latitude ( $\beta = -0.003303$ , 95% CI = [-0.004121, -0.002484], **Table 5, Figure 6**). The negative slope of this relationship indicates that change in NDVI is greater at lower latitudes, indicating more treeline greenness in the Tropics and Subtropics in more recent years.

| Term                         | Estimate             | SE                  | Lower 95% CI         | Upper 95% CI        |
|------------------------------|----------------------|---------------------|----------------------|---------------------|
| Intercept                    | $1.2 	imes 10^{-1}$  | $1.9\times10^{-2}$  | $8.6 	imes 10^{-2}$  | $1.6 	imes 10^{-1}$ |
| Latitude                     | $-3.3\times10^{-3}$  | $4.2 	imes 10^{-4}$ | $-4.1 	imes 10^{-3}$ | $-2.5\times10^{-3}$ |
| Random intercept (variance)  | 3.7                  |                     |                      |                     |
| Random intercept (variance)  | $2.4 \times 10^{-3}$ |                     |                      |                     |
| Random intercept (std. dev.) | $4.9 \times 10^{-2}$ |                     |                      |                     |
| Residual (variance)          | $1.5 \times 10^{-3}$ |                     |                      |                     |
| Residual (std. dev.)         | $3.9 \times 10^{-2}$ |                     |                      |                     |

**Table 5.** Model summary of the top linear mixed model. Fixed and random effect outputs are shown. The response variable for this model was change in treeline NDVI. Significance is denoted by bold text, and was assessed by observing whether or not the confidence interval overlapped zero.

Among the set of spatial mixed models, the top model concurred with the top linear mixed model. Latitude was again significantly negatively related to change in treeline NDVI ( $\beta$  = -0.003852, 95% CI = [-0.005047, -0.002705], **Table 6, Figure 6**); no other variables had significant effects. The negative slope underlines the linkage between lower latitudes and more intense treeline movement. This model was the best performing overall out of all models tested that had change in NDVI as the response variable. The smoothness parameter ( $\nu$ ) was estimated at 0.259, indicating moderate spatial continuity in the data. The range parameter ( $\rho$ ) was 1.006, suggesting that spatial correlation between observations diminishes rapidly over very short distances. The variance of the spatial random effect ( $\lambda$ ) was estimated at 0.00298, reflecting residual spatial variability in the data. The residual variance ( $\phi$ ) was estimated at 0.00107, representing the remaining variability not explained by the spatial random effect (**Table 6**).

| Term                         | Estimate             | SE                  | T-Value | Lower 95% CI       | Upper 95% CI        |
|------------------------------|----------------------|---------------------|---------|--------------------|---------------------|
| Intercept                    | $1.4 	imes 10^{-1}$  | $2.4 	imes 10^{-2}$ | 5.7     | $9.1\times10^{-2}$ | $1.9\times10^{-1}$  |
| Latitude                     | $-3.9\times10^{-3}$  | $5.8 	imes 10^{-4}$ | -6.6    | $-5.0	imes10^{-3}$ | $-2.7\times10^{-3}$ |
| Random intercept (variance)  | $3.0 \times 10^{-3}$ |                     |         |                    |                     |
| Random intercept (std. dev.) | $5.5 \times 10^{-2}$ |                     |         |                    |                     |
| Residual (variance)          | $1.1 \times 10^{-3}$ |                     |         |                    |                     |
| Residual (std. dev.)         | $3.3 \times 10^{-2}$ |                     |         |                    |                     |

**Table 6.** Model summary of the top spatial mixed model. Fixed and random effect outputs are shown. The response variable for this model was change in treeline NDVI. Significance is denoted by bold text, and was assessed by observing whether or not the confidence interval overlapped zero.

**Figure 6.** Summary of the univariate relationship between NDVI at manually identified 2017 treeline elevations. Change in NDVI, on the y-axis, represents 2017 NDVI - 1984 NDVI. Latitude, which was significant in the models with change in treeline NDVI as the response variable, is shown on the x-axis. The regression line from the linear model of change in treeline NDVI versus latitude is denoted by the black line.

# 238 4 Discussion

## 4.1 Overview

This study represents a first broad-scope view of spatial patterns of temporal shifts in treeline elevation across a major world region. In that sense, it is novel, but our insights have been limited by a number of data-related challenges: e.g., the necessity of eliminating the northernmost set of high peaks because treelines were not uniquely associated with individual peaks, as well as the removal of a number of peaks from consideration owing to positive slopes in regression models relating NDVI to elevation. These complications point out the nascent nature of this endeavor, and the need for quite a bit more exploration and experimentation.

Our results underlined some previous results, such as treelines occurring at higher elevations in the Tropics and Subtropics, and at lower elevations at higher latitudes (Körner, 1998). This broad pattern makes sense, of course, if one thinks of the conditions present at the highest elevations—they are at the extremes of what is survivable for upright trees (Körner, 2021). If treeline is set at least in part by hard physiological limits, and given global climate patterns and how they vary with latitude (Peterson et al., 2016), then high-latitude treelines would necessarily be lower in elevation.

More importantly and more novel, however, our results show clear associations between magnitude of treeline shift and latitude, such that tropical treelines have shifted upward faster than higher-latitude treelines in recent decades (Jiménez-García et al., 2021). Such an effect has not been appreciated or reported previously, at least to our knowledge, but may relate to the greater physiological flexibility that may characterize tropical treelines: that is, high-latitude treelines may be fixed in elevation by hard physiological limits related to freeze tolerance (Körner, 2021). This focus of treeline mobility in the tropical zone, unfortunately, coincides with significant knowledge gaps, given that the great majority of detailed studies of treelines and their dynamics has been conducted on peaks at higher latitudes (Shi et al., 2022; Zhao et al., 2015; Körner, 1998; Lu et al., 2021).

Our results were suggestive of further effects, related to longitude and perhaps distance to coastlines; proximity to ocean has been underlined in past studies as important in determining treeline elevations at least (Hansson et al., 2023). That is, although we included a variable summarizing geographic distance to coastline, it did not have any significant effect in the best models. Rather, in some of the models that ranked among the best, effects of longitude were indeed substantial. We suspect that this lack of clear effect of distance to coastlines may be related to the relatively minor representation of peaks close to coastlines in our dataset.

# 4.2 Limitations

The deepest concern regarding the analyses presented herein is, of course, the relatively short time span covered by the Landsat imagery, with our analyses spanning just a bit more than three decades. This time span is, of course, what is available from remote-sensing data streams, as Landsat is among the deepest-time remote-sensing data sources available anywhere. Even our relatively short time span of Landsat data, however, does cross the use of multiple sensors to produce the imagery, which may introduce noise into the results that we present herein (Vogelmann et al., 2016). The only remedy to this concern about time

span is therefore to appeal to other data sources, such as aerial or ground-based photos (Jiménez-García et al., 2021; Peterson 270 271 et al., 2022).

This study covered an impressive expanse in western North America, from 9.4°N in Costa Rica north to 54.1°N in southwestern Canada, and from the shores of the Pacific Ocean to the Front Range of the Rocky Mountains in Colorado. However, this geographic span includes relatively fewer high mountain peaks in Mexico and Central America, at least compared with 275 the northern peaks in the study; a further possible limitation of our work stems from the broad latitudinal gaps in northern Mexico. Finally, our inability to associate specific treelines with specific high peaks north of southernmost Canada meant that 276 the highest-latitude peaks could not be included in the study. Some of these concerns can be remedied by broadening the area of study and analysis still further, perhaps globally, but the latter concern will remain complicated, as very high latitude peaks tend to be mostly above treeline, such that we do not see a way to create a peak-based analysis of those regions.

Finally, a concern could be that of anthropogenic effects that are not related to climate. That is, although we eliminated from consideration any peaks that had human activities visible at the peak or near treeline (e.g., agricultural activities), we could not control for changing practices of fire control, for example. In this sense, if fire control has been implemented or has become more effective over the past few decades, that—unrelated to climate—could elevate NDVI owing to reduced fire-based removal of vegetation. We hope that the broad variety of peaks included in this study will avoid any confounding effects of this concern.

#### 4.3 Conclusions and Next Steps

279

- The results of this study point rather dramatically to the crucial importance of a major knowledge gap regarding high-elevation 287 vegetation dynamics. That is, the bias of treeline studies away from tropical regions and towards temperate-zone and borealzone regions coincides—unfortunately—with the most dramatic regions of treeline elevational shifts. As we have pointed out 288 289 in previous contributions (Jiménez-García et al., 2021), treelines in the Tropics and their dynamics remain little-documented and poorly characterized. 290
- At the same time, the results of this study and others (Peterson et al., 2022; Singh et al., 2012) indicate that remote-sensing 292 data streams are both relevant and informative, and have as a result been incorporated into many treeline studies (Garbarino et al., 2023). Although the detail available in on-the-ground studies cannot be achieved, significant insight can indeed be gained 293 from satellite-based observations and data streams, particularly when multiple data streams are integrated (Garbarino et al., 294 2023). As such, we are in the process of extending this approach globally and using more-diverse remote sensing data streams, 295 296 in the hope of garnering additional useful insights into patterns of treeline change worldwide, and into processes that drive treeline change phenomena. 297
- Code and data availability. All data and code are available on a public Github repository found at the following URL: 298
- https://github.com/jocori/GeographicTreelinePatterns.git

- Author contributions. JLC cleaned and organized the dataset for analysis, prepared the response variables, and conducted the data analysis.

  She also created figures and tables, wrote the methods, results, and literature review sections, edited the introduction, discussion, and abstract, and formatted the manuscript for publication in LaTeX.
- DJG helped to design the study, developed the sampling protocol, and executed the extraction of the NDVI data from satellite imagery.

  He also helped to guide the data analysis, and to improve the text of the manuscript.
- XL helped to design the study and assisted with scripts for data extraction and preparation
- ATP helped to design the study, as well as the data analysis. He performed key manual data refinement steps in identifying treeline and quality-controlling transects. He assisted with the design of the figures and editing and improvement of the text of the manuscript.
- Competing interests. The authors declare that they have no conflict of interest.
- *Acknowledgements*. We thank Andrés Lira Noriega for his valuable contributions during the early stages of data analysis, and Jeffrey Munroe 310 for insight into treeline biology generally. Additionally, we acknowledge the University of Kansas Office of the Provost for funding support.

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

# 403 Appendix A: Supplementary Tables

| Model Terms                                                                              | AIC                 | Delta AIC          | Weight                |
|------------------------------------------------------------------------------------------|---------------------|--------------------|-----------------------|
| # Stations After Treeline + Direction + Latitude x Longitude                             | $6.8 \times 10^{3}$ | 0.000              | $5.7 \times 10^{-1}$  |
| Direction + Latitude x Longitude                                                         | $6.8 \times 10^3$   | $7.0\times10^{-1}$ | $4.0 \times 10^{-1}$  |
| Latitude + Longitude + # Stations After Treeline + Direction                             | $6.8 \times 10^3$   | 7.4                | $1.4 \times 10^{-2}$  |
| Latitude + Longitude + Direction                                                         | $6.8 \times 10^3$   | 8.2                | $9.7\times10^{-3}$    |
| Distance to the Coast (m) + Direction + # Stations After Treeline + Latitude x Longitude | $6.8 \times 10^3$   | $1.7\times10^{1}$  | $9.7\times10^{-5}$    |
| Distance to the Coast (m) + Direction + Latitude x Longitude                             | $6.8 \times 10^3$   | $1.8\times10^{1}$  | $6.8 \times 10^{-5}$  |
| Distance to the Coast (m) + Direction + # Stations After Treeline + Latitude + Longitude | $6.8 \times 10^3$   | $1.9\times10^{1}$  | $4.8 \times 10^{-5}$  |
| Distance to the Coast (m) + Direction + Latitude + Longitude                             | $6.8 \times 10^3$   | $1.9\times10^{1}$  | $3.4\times10^{-5}$    |
| # Stations After Treeline + Latitude x Longitude                                         | $6.9 \times 10^3$   | $5.6 \times 10^1$  | $4.7 \times 10^{-13}$ |
| Latitude x Longitude                                                                     | $6.9 \times 10^3$   | $5.6\times10^{1}$  | $3.5 \times 10^{-13}$ |
| Latitude + Longitude + # Stations After Treeline                                         | $6.9 \times 10^3$   | $6.4 \times 10^1$  | $9.2 \times 10^{-15}$ |
| Latitude + Longitude                                                                     | $6.9 \times 10^3$   | $6.4 \times 10^1$  | $6.8 \times 10^{-15}$ |
| Latitude                                                                                 | $6.9 \times 10^3$   | $6.9 \times 10^1$  | $7.3 \times 10^{-16}$ |
| Longitude                                                                                | $6.9 \times 10^3$   | $7.1\times10^{1}$  | $2.2\times10^{-16}$   |
| Distance to the Coast (m) + # Stations After Treeline + Latitude x Longitude             | $6.9 \times 10^3$   | $7.3 \times 10^1$  | $7.9 \times 10^{-17}$ |
| Distance to the Coast (m) + Latitude x Longitude                                         | $6.9 \times 10^3$   | $7.4 \times 10^1$  | $5.9 \times 10^{-17}$ |
| Distance to the Coast (m) + # Stations After Treeline + Latitude + Longitude             | $6.9 \times 10^3$   | $7.4\times10^{1}$  | $3.8\times10^{-17}$   |
| Latitude + Longitude + Distance to the Coast (m)                                         | $6.9 \times 10^3$   | $7.5\times10^{1}$  | $2.9\times10^{-17}$   |

**Table A1.** AIC table comparing all linear mixed models which had change in treeline elevation as the response variable. In all, 18 models were involved in this comparison.

| Term                                                                                               | AIC                 | Delta AIC         | Weight               |
|----------------------------------------------------------------------------------------------------|---------------------|-------------------|----------------------|
| Latitude                                                                                           | $6.9 \times 10^{3}$ | 0.000             | $3.5 \times 10^{-1}$ |
| Latitude × Longitude                                                                               | $6.9 \times 10^3$   | 1.2               | $1.9\times10^{-1}$   |
| Longitude                                                                                          | $6.9 \times 10^3$   | 1.5               | $1.6\times10^{-1}$   |
| # Stations After Treeline + Latitude × Longitude                                                   | $6.9 \times 10^3$   | 3.0               | $7.6\times10^{-2}$   |
| Latitude + Longitude                                                                               | $6.9 \times 10^3$   | 3.2               | $7.1\times10^{-2}$   |
| Distance to the Coast (m) + Latitude $\times$ Longitude                                            | $6.9 \times 10^3$   | 3.9               | $4.9\times10^{-2}$   |
| Latitude + Longitude + Distance to the Coast (m)                                                   | $6.9 \times 10^3$   | 4.5               | $3.6\times10^{-2}$   |
| Latitude + Longitude + # Stations After Treeline                                                   | $6.9 \times 10^3$   | 5.1               | $2.7\times10^{-2}$   |
| Distance to the Coast (m) + $\#$ Stations After Treeline + Latitude $\times$ Longitude             | $6.9 \times 10^3$   | 5.8               | $1.9\times10^{-2}$   |
| Distance to the Coast (m) + # Stations After Treeline + Latitude + Longitude                       | $6.9 \times 10^3$   | 6.4               | $1.4\times10^{-2}$   |
| Direction + Latitude × Longitude                                                                   | $6.9 \times 10^3$   | $1.0\times10^{1}$ | $1.8\times10^{-3}$   |
| # Stations After Treeline + Direction + Latitude × Longitude                                       | $6.9 \times 10^3$   | $1.2\times10^{1}$ | $7.4\times10^{-4}$   |
| Latitude + Longitude + Direction                                                                   | $6.9 \times 10^3$   | $1.2 \times 10^1$ | $6.9\times10^{-4}$   |
| Distance to the Coast (m) + Direction + Latitude $\times$ Longitude                                | $6.9 \times 10^3$   | $1.3 \times 10^1$ | $4.6\times10^{-4}$   |
| Distance to the Coast (m) + Direction + Latitude + Longitude                                       | $6.9 \times 10^3$   | $1.4\times10^{1}$ | $3.4\times10^{-4}$   |
| Latitude + Longitude + # Stations After Treeline + Direction                                       | $6.9 \times 10^3$   | $1.4\times10^{1}$ | $2.7\times10^{-4}$   |
| Distance to the Coast (m) + Direction + $\#$ Stations After Treeline + Latitude $\times$ Longitude | $6.9\times10^3$     | $1.5\times10^{1}$ | $1.9\times10^{-4}$   |
| Distance to the Coast (m) + Direction + # Stations After Treeline + Latitude + Longitude           | $6.9\times10^3$     | $1.6\times10^{1}$ | $1.4\times10^{-4}$   |

**Table A2.** AIC table comparing all spatial mixed models which had change in treeline elevation as the response variable. In all, 18 models were involved in this comparison.

| Model Type | Terms                                                                                    | AIC               | Delta AIC           | Weight               |
|------------|------------------------------------------------------------------------------------------|-------------------|---------------------|----------------------|
| Spatial    | Latitude                                                                                 | $6.9 \times 10^3$ | 0.000               | $3.4 \times 10^{-1}$ |
| Spatial    | Direction + Latitude $\times$ Longitude                                                  | $6.9 \times 10^3$ | $1.0 \times 10^1$   | $1.8 \times 10^{-3}$ |
| Linear     | # Stations After Treeline + Latitude $\times$ Longitude                                  | $6.9 \times 10^3$ | $1.1 \times 10^1$   | $1.4 \times 10^{-3}$ |
| Linear     | Distance to the Coast (m) + Latitude $\times$ Longitude                                  | $6.9 \times 10^3$ | $1.1\times10^{1}$   | $1.3 \times 10^{-3}$ |
| Spatial    | Latitude × Longitude                                                                     | $6.9 \times 10^3$ | 1.2                 | $1.9 \times 10^{-1}$ |
| Linear     | Latitude + Longitude + Distance to the Coast (m)                                         | $6.9\times10^3$   | $1.2\times10^{1}$   | $7.5 \times 10^{-4}$ |
| Spatial    | # Stations After Treeline + Direction + Latitude × Longitude                             | $6.9 \times 10^3$ | $1.2 \times 10^1$   | $7.3 \times 10^{-4}$ |
| Spatial    | Latitude + Longitude + Direction                                                         | $6.9\times10^3$   | $1.2\times10^{1}$   | $6.8 \times 10^{-4}$ |
| Linear     | Distance to the Coast (m) + $\#$ Stations After Treeline + Latitude $\times$ Longitude   | $6.9 \times 10^3$ | $1.3 \times 10^1$   | $5.0 \times 10^{-4}$ |
| Spatial    | Distance to the Coast (m) + Direction + Latitude × Longitude                             | $6.9 \times 10^3$ | $1.3 \times 10^1$   | $4.6 \times 10^{-4}$ |
| Spatial    | Distance to the Coast (m) + Direction + Latitude + Longitude                             | $6.9\times10^3$   | $1.4\times10^{1}$   | $3.4 \times 10^{-4}$ |
| Linear     | Distance to the Coast (m) + # Stations After Treeline + Latitude + Longitude             | $6.9 \times 10^3$ | $1.4\times10^{1}$   | $2.9 \times 10^{-4}$ |
| Spatial    | Latitude + Longitude + # Stations After Treeline + Direction                             | $6.9 \times 10^3$ | $1.4 \times 10^1$   | $2.7 \times 10^{-4}$ |
| Spatial    | Longitude                                                                                | $6.9\times10^3$   | 1.5                 | $1.6 \times 10^{-1}$ |
| Spatial    | Distance to the Coast (m) + Direction + # Stations After Treeline + Latitude × Longitude | $6.9\times10^3$   | $1.5\times10^{1}$   | $1.8 \times 10^{-4}$ |
| Spatial    | Distance to the Coast (m) + Direction + # Stations After Treeline + Latitude + Longitude | $6.9\times10^3$   | $1.6\times10^{1}$   | $1.4 \times 10^{-4}$ |
| Linear     | Latitude                                                                                 | $6.9\times10^3$   | $1.7\times10^{1}$   | $8.1 \times 10^{-5}$ |
| Linear     | Latitude + Longitude                                                                     | $6.9\times10^3$   | $1.8\times10^{1}$   | $3.8 \times 10^{-5}$ |
| Linear     | Direction + Latitude × Longitude                                                         | $6.9 \times 10^3$ | $1.9 \times 10^{1}$ | $3.2 \times 10^{-5}$ |
| Linear     | Longitude                                                                                | $6.9\times10^3$   | $1.9\times10^{1}$   | $2.0 \times 10^{-5}$ |
| Linear     | Latitude + Longitude + # Stations After Treeline                                         | $6.9\times10^3$   | $2.0 \times 10^{1}$ | $1.5 \times 10^{-5}$ |
| Linear     | # Stations After Treeline + Direction + Latitude × Longitude                             | $6.9\times10^3$   | $2.0 \times 10^{1}$ | $1.3 \times 10^{-5}$ |
| Linear     | Distance to the Coast (m) + Direction + Latitude × Longitude                             | $6.9\times10^3$   | $2.1 \times 10^1$   | $1.2 \times 10^{-5}$ |
| Linear     | Distance to the Coast (m) + Direction + Latitude + Longitude                             | $6.9\times10^3$   | $2.2 \times 10^{1}$ | $7.0 \times 10^{-6}$ |
| Linear     | Distance to the Coast (m) + Direction + # Stations After Treeline + Latitude × Longitude | $6.9 \times 10^3$ | $2.2 \times 10^1$   | $4.8 \times 10^{-6}$ |
| Linear     | Distance to the Coast (m) + Direction + # Stations After Treeline + Latitude + Longitude | $6.9\times10^3$   | $2.3 \times 10^1$   | $2.8 \times 10^{-6}$ |
| Linear     | Latitude + Longitude + Direction                                                         | $6.9 \times 10^3$ | $2.7\times10^{1}$   | $4.3 \times 10^{-7}$ |
| Linear     | Latitude + Longitude + # Stations After Treeline + Direction                             | $6.9 \times 10^3$ | $2.9 \times 10^{1}$ | $1.8 \times 10^{-7}$ |
| Spatial    | # Stations After Treeline + Latitude × Longitude                                         | $6.9\times10^3$   | 3.0                 | $7.5 \times 10^{-2}$ |
| Spatial    | Latitude + Longitude                                                                     | $6.9\times10^3$   | 3.2                 | $7.0 \times 10^{-2}$ |
| Spatial    | Distance to the Coast (m) + Latitude $\times$ Longitude                                  | $6.9 \times 10^3$ | 3.9                 | $4.9 \times 10^{-2}$ |
| Spatial    | Latitude + Longitude + Distance to the Coast (m)                                         | $6.9\times10^3$   | 4.5                 | $3.6 \times 10^{-2}$ |
| Spatial    | Latitude + Longitude + # Stations After Treeline                                         | $6.9\times10^3$   | 5.1                 | $2.7 \times 10^{-2}$ |
| Spatial    | Distance to the Coast (m) + # Stations After Treeline + Latitude × Longitude             | $6.9\times10^3$   | 5.8                 | $1.9 \times 10^{-2}$ |
| Spatial    | Distance to the Coast (m) + # Stations After Treeline + Latitude + Longitude             | $6.9\times10^3$   | 6.4                 | $1.4 \times 10^{-2}$ |
| Linear     | Latitude × Longitude                                                                     | $6.9\times10^3$   | 9.2                 | $3.5 \times 10^{-3}$ |

**Table A3.** AIC table comparing all linear mixed models and spatial mixed models which had change in treeline elevation as the response variable. In all, 36 models were involved in this comparison.

| Terms                                                                                    | AIC                | Delta AIC           | Weight                |
|------------------------------------------------------------------------------------------|--------------------|---------------------|-----------------------|
| Latitude                                                                                 | $-1.5 \times 10^3$ | 0.00                | $7.4 \times 10^{-1}$  |
| Longitude                                                                                | $-1.5 \times 10^3$ | 2.1                 | $2.5\times10^{-1}$    |
| Latitude + Longitude                                                                     | $-1.5 \times 10^3$ | $1.2\times10^{1}$   | $1.9\times10^{-3}$    |
| Latitude + Longitude + # Stations After Treeline                                         | $-1.5 \times 10^3$ | $2.9 \times 10^{1}$ | $3.8\times10^{-7}$    |
| Latitude × Longitude                                                                     | $-1.5 \times 10^3$ | $3.2\times10^{1}$   | $9.9\times10^{-8}$    |
| Latitude + Longitude + Distance to the Coast (m)                                         | $-1.4 \times 10^3$ | $4.6\times10^{1}$   | $8.0\times10^{-11}$   |
| # Stations After Treeline + Latitude × Longitude                                         | $-1.4 \times 10^3$ | $4.9 \times 10^{1}$ | $2.0\times10^{-11}$   |
| Distance to the Coast (m) + # Stations After Treeline + Latitude + Longitude             | $-1.4 \times 10^3$ | $6.3\times10^{1}$   | $1.6 \times 10^{-14}$ |
| Distance to the Coast $(m)$ + Latitude $\times$ Longitude                                | $-1.4 \times 10^3$ | $6.4 \times 10^1$   | $9.4\times10^{-15}$   |
| Latitude + Longitude + Direction                                                         | $-1.4 \times 10^3$ | $7.4\times10^{1}$   | $7.0\times10^{-17}$   |
| Distance to the Coast (m) + # Stations After Treeline + Latitude × Longitude             | $-1.4 \times 10^3$ | $8.1\times10^{1}$   | $1.9\times10^{-18}$   |
| Latitude + Longitude + # Stations After Treeline + Direction                             | $-1.4 \times 10^3$ | $9.1\times10^{1}$   | $1.5\times10^{-20}$   |
| Direction + Latitude × Longitude                                                         | $-1.4 \times 10^3$ | $9.4 \times 10^1$   | $3.3\times10^{-21}$   |
| Distance to the Coast (m) + Direction + Latitude + Longitude                             | $-1.4 \times 10^3$ | $1.1\times10^2$     | $2.5\times10^{-24}$   |
| # Stations After Treeline + Direction + Latitude × Longitude                             | $-1.4 \times 10^3$ | $1.1\times10^2$     | $7.1 \times 10^{-25}$ |
| Distance to the Coast (m) + Direction + # Stations After Treeline + Latitude + Longitude | $-1.4 \times 10^3$ | $1.3\times10^2$     | $5.3\times10^{-28}$   |
| Distance to the Coast (m) + Direction + Latitude × Longitude                             | $-1.4 \times 10^3$ | $1.3\times10^2$     | $3.0\times10^{-28}$   |
| Distance to the Coast (m) + Direction + # Stations After Treeline + Latitude × Longitude | $-1.4 \times 10^3$ | $1.4 \times 10^2$   | $6.3\times10^{-32}$   |

**Table A4.** AIC table comparing all linear mixed models which had change in treeline NDVI as the response variable. In all, 18 models were involved in this comparison.

| Terms                                                                                    | AIC                | Delta AIC | Weight               |
|------------------------------------------------------------------------------------------|--------------------|-----------|----------------------|
| Latitude                                                                                 | $-1.5 \times 10^3$ | 0.000     | $4.0 \times 10^{-1}$ |
| Latitude + Longitude                                                                     | $-1.5 \times 10^3$ | 1.7       | $1.8\times10^{-1}$   |
| Latitude × Longitude                                                                     | $-1.5 \times 10^3$ | 3.6       | $6.6\times10^{-2}$   |
| Latitude + Longitude + # Stations After Treeline                                         | $-1.5 \times 10^3$ | 3.6       | $6.5\times10^{-2}$   |
| Latitude + Longitude + Distance to the Coast (m)                                         | $-1.5 \times 10^3$ | 3.0       | $8.9\times10^{-2}$   |
| Distance to the Coast (m) + Latitude $\times$ Longitude                                  | $-1.5 \times 10^3$ | 4.8       | $3.7\times10^{-2}$   |
| Longitude                                                                                | $-1.5 \times 10^3$ | 4.6       | $4.1\times10^{-2}$   |
| # Stations After Treeline + Latitude × Longitude                                         | $-1.5 \times 10^3$ | 5.6       | $2.5\times10^{-2}$   |
| Distance to the Coast (m) + # Stations After Treeline + Latitude + Longitude             | $-1.5 \times 10^3$ | 5.0       | $3.3\times10^{-2}$   |
| Latitude + Longitude + Direction                                                         | $-1.5 \times 10^3$ | 6.2       | $1.8\times10^{-2}$   |
| Distance to the Coast (m) + # Stations After Treeline + Latitude $\times$ Longitude      | $-1.5 \times 10^3$ | 6.8       | $1.4\times10^{-2}$   |
| Distance to the Coast (m) + Direction + Latitude + Longitude                             | $-1.5 \times 10^3$ | 7.7       | $8.5\times10^{-3}$   |
| Direction + Latitude × Longitude                                                         | $-1.5 \times 10^3$ | 8.2       | $6.6\times10^{-3}$   |
| Latitude + Longitude + # Stations After Treeline + Direction                             | $-1.5 \times 10^3$ | 8.1       | $7.1\times10^{-3}$   |
| Distance to the Coast (m) + Direction + Latitude $\times$ Longitude                      | $-1.5 \times 10^3$ | 9.5       | $3.6\times10^{-3}$   |
| Distance to the Coast (m) + Direction + # Stations After Treeline + Latitude + Longitude | $-1.5\times10^3$   | 9.6       | $3.3\times10^{-3}$   |

**Table A5.** AIC table comparing all spatial mixed models which had change in treeline NDVI as the response variable. In all, 18 models were involved in this comparison.

| Model Type | Terms                                                               | AIC                  | Delta AIC           | Weight               |
|------------|---------------------------------------------------------------------|----------------------|---------------------|----------------------|
| Spatial    | Latitude                                                            | $-1.5 \times 10^{3}$ | 0.000               | $4.0 \times 10^{-1}$ |
| Spatial    | # Stations After Treeline + Direction + Latitude $\times$ Longitude | $-1.5 \times 10^{3}$ | $1.0 \times 10^{1}$ | $2.6 \times 10^{-3}$ |
| Spatial    | Distance to the Coast (m) + Direction + # Stations After Tree-      | $-1.5 \times 10^{3}$ | $1.1 \times 10^{1}$ | $1.4 \times 10^{-3}$ |
|            | line + Latitude × Longitude                                         |                      |                     |                      |
| Linear     | Latitude + Longitude + Distance to the Coast (m)                    | $-1.5 \times 10^{3}$ | $1.5 \times 10^{1}$ | $2.1 \times 10^{-4}$ |
| Linear     | Latitude                                                            | $-1.5 \times 10^{3}$ | $1.5 \times 10^{1}$ | $2.1 \times 10^{-4}$ |
| Linear     | Latitude + Longitude                                                | $-1.5 \times 10^{3}$ | $1.5 \times 10^{1}$ | $1.8 \times 10^{-4}$ |
| Linear     | Distance to the Coast (m) + Latitude $\times$ Longitude             | $-1.5 \times 10^{3}$ | $1.6 \times 10^{1}$ | $1.3 \times 10^{-4}$ |
| Spatial    | Latitude + Longitude                                                | $-1.5 \times 10^{3}$ | 1.7                 | $1.8 \times 10^{-1}$ |
| Linear     | Latitude × Longitude                                                | $-1.5 \times 10^{3}$ | $1.7 \times 10^{1}$ | $9.3 \times 10^{-5}$ |
| Linear     | Distance to the Coast (m) + # Stations After Treeline + Lati-       | $-1.5 \times 10^{3}$ | $1.7 \times 10^{1}$ | $8.0 \times 10^{-5}$ |
|            | tude + Longitude                                                    |                      |                     |                      |
| Linear     | Latitude + Longitude + # Stations After Treeline                    | $-1.5 \times 10^{3}$ | $1.7 \times 10^{1}$ | $6.9 \times 10^{-5}$ |
| Linear     | Longitude                                                           | $-1.5 \times 10^{3}$ | $1.8 \times 10^{1}$ | $6.0 \times 10^{-5}$ |
| Linear     | Distance to the Coast (m) + # Stations After Treeline + Lati-       | $-1.5 \times 10^{3}$ | $1.8 \times 10^{1}$ | $4.8 \times 10^{-5}$ |
|            | tude × Longitude                                                    |                      |                     |                      |
| Linear     | # Stations After Treeline + Latitude × Longitude                    | $-1.5 \times 10^{3}$ | $1.9 \times 10^{1}$ | $3.5 \times 10^{-5}$ |
| Linear     | Latitude + Longitude + Direction                                    | $-1.5 \times 10^{3}$ | $1.9 \times 10^{1}$ | $3.2 \times 10^{-5}$ |
| Linear     | Distance to the Coast (m) + Direction + Latitude + Longitude        | $-1.5 \times 10^{3}$ | $1.9 \times 10^{1}$ | $3.2 \times 10^{-5}$ |
| Linear     | Distance to the Coast (m) + Direction + Latitude $\times$ Longitude | $-1.5 \times 10^{3}$ | $2.0 \times 10^{1}$ | $1.9 \times 10^{-5}$ |
| Linear     | Direction + Latitude × Longitude                                    | $-1.5 \times 10^{3}$ | $2.0 \times 10^{1}$ | $1.5 \times 10^{-5}$ |
| Linear     | Latitude + Longitude + # Stations After Treeline + Direction        | $-1.5 \times 10^{3}$ | $2.1 \times 10^{1}$ | $1.3 \times 10^{-5}$ |
| Linear     | Distance to the Coast (m) + Direction + # Stations After Tree-      | $-1.5 \times 10^{3}$ | $2.1 \times 10^1$   | $1.3 \times 10^{-5}$ |
|            | line + Latitude + Longitude                                         |                      |                     |                      |
| Linear     | Distance to the Coast (m) + Direction + # Stations After Tree-      | $-1.5 \times 10^{3}$ | $2.2 \times 10^{1}$ | $7.8 \times 10^{-6}$ |
|            | line + Latitude × Longitude                                         |                      |                     |                      |
| Linear     | # Stations After Treeline + Direction + Latitude × Longitude        | $-1.5 \times 10^{3}$ | $2.2 \times 10^{1}$ | $6.0 \times 10^{-6}$ |
| Spatial    | Latitude + Longitude + Distance to the Coast (m)                    | $-1.5 \times 10^{3}$ | 3.0                 | $8.9 \times 10^{-2}$ |
| Spatial    | Latitude × Longitude                                                | $-1.5 \times 10^{3}$ | 3.6                 | $6.6 \times 10^{-2}$ |
| Spatial    | Latitude + Longitude + # Stations After Treeline                    | $-1.5 \times 10^{3}$ | 3.6                 | $6.5 \times 10^{-2}$ |
| Spatial    | Longitude                                                           | $-1.5 \times 10^{3}$ | 4.6                 | $4.1 \times 10^{-2}$ |
| Spatial    | Distance to the Coast (m) + Latitude × Longitude                    | $-1.5 \times 10^{3}$ | 4.8                 | $3.7 \times 10^{-2}$ |
| Spatial    | Distance to the Coast (m) + # Stations After Treeline + Lati-       | $-1.5 \times 10^{3}$ | 5.0                 | $3.3 \times 10^{-2}$ |
|            | tude + Longitude                                                    |                      |                     |                      |
| Spatial    | # Stations After Treeline + Latitude × Longitude                    | $-1.5 \times 10^{3}$ | 5.6                 | $2.5 \times 10^{-2}$ |
| Spatial    | Latitude + Longitude + Direction                                    | $-1.5 \times 10^{3}$ | 6.2                 | $1.8 \times 10^{-2}$ |
| Spatial    | Distance to the Coast (m) + # Stations After Treeline + Lati-       | $-1.5 \times 10^{3}$ | 6.8                 | $1.4 \times 10^{-2}$ |
|            | tude × Longitude                                                    |                      |                     |                      |
| Spatial    | Distance to the Coast (m) + Direction + Latitude + Longitude        | $-1.5 \times 10^{3}$ | 7.7                 | $8.5 \times 10^{-3}$ |
| Spatial    | Latitude + Longitude + # Stations After Treeline + Direction        | $-1.5 \times 10^{3}$ | 8.1                 | $7.0 \times 10^{-3}$ |
| Spatial    | Direction + Latitude × Longitude                                    | $-1.5 \times 10^{3}$ | 8.2                 | $6.6 \times 10^{-3}$ |
| Spatial    | Distance to the Coast (m) + Direction + Latitude × Longitude        | $-1.5 \times 10^{3}$ | 9.5                 | $3.6 \times 10^{-3}$ |
| Spatial    | Distance to the Coast (m) + Direction + # Stations After Tree-      | $-1.5 \times 10^{3}$ | 9.6                 | $3.3 \times 10^{-3}$ |
| 1          | line + Latitude + Longitude                                         |                      | ~-~                 | 0.0 20               |

**Table A6.** AIC table comparing all linear mixed models and spatial mixed models which had change in treeline NDVI as the response variable. In all, 36 models were involved in this comparison.