# Peer review of "Geographic patterns of upward shifts in treeline vegetation across western North America, 1984–2017"

_EGUsphere, 2025_

## Author Response (AR1)

**Response to Editors**

We thank the Associate Editor, Dr. Matteo Garbarino and the Co-editor-in-chief, Dr. Frank Hagedorn for their careful evaluation of our manuscript and for recognizing the improvements made during revision. We have reviewed and adjusted the manuscript text and figures to ensure compliance with the formatting guidelines of *Biogeosciences* by using the LaTeX template provided by the journal. All minor revisions requested by the editors have been implemented, including thorough formatting checks of the manuscript content and visual materials.

Below, we provide a detailed point-by-point response to all comments from the peer reviewers, including a summary of the specific changes made in the manuscript.

Sincerely,

Joanna Corimanya, Daniel Jiménez-García, Xingong Li, and Townsend Peterson

**Referee #1:**

This manuscript presents a novel, well-structured remote sensing analysis of changes in treeline elevations across a vast latitudinal gradient in western North America. The integration of a large-scale spatial design, hierarchical regression modeling, and clear documentation of methods adds valuable insight into geographic variation in treeline response to climate change. The focus on underrepresented tropical and subtropical peaks is particularly commendable. However, I have several comments, which may help to improve this manuscript.

We appreciate the reviewer's positive comments. We have indeed attempted to integrate diverse methodologies and document each and all of them carefully with explicit documentation and—where possible—coding.

My main concern stems from sensor harmonization and NDVI consistency. The study integrates Landsat 5, 7, and 8, which have known differences in NDVI due to varying spectral response functions (especially between Landsat 8 and earlier sensors, Roy et al., 2016). Please clarify how you accounted for cross-sensor calibration. Without harmonization, inter-sensor variability could bias estimates of NDVI-derived treeline elevation shifts.

We recognize the reviewer's concern regarding sensor harmonization and NDVI consistency across Landsat 5, 7, and 8. To address this, we relied on the pre-processed LANDSAT\_LT5\_L1T\_ANNUAL\_NDVI collection in Google Earth Engine, which applies cross-sensor calibration to ensure consistency among different Landsat platforms (Roy et al., 2016). Although this GEE collection is now deprecated, its underlying methodology—described in Roy et al. (2016) and detailed at <a href="https://www.sciencedirect.com/science/article/abs/pii/S0034425709000169">https://www.sciencedirect.com/science/article/abs/pii/S0034425709000169</a>—accounts for spectral response differences and normalizes NDVI values across sensors. For full transparency, we have made our processing scripts available so that users can replicate the harmonization steps or compute annual NDVI composites directly from Level-1 terrain-corrected reflectance data. These scripts implement the same calibration

routines used by the original collection, thereby minimizing any inter-sensor bias in our NDVI-derived estimates of treeline elevation shifts.

Fig. 1: The inclusion of eastern North America may be visually misleading since it is not studied. I suggest the authors to limit the extent of the map to only regions with analyzed peaks and remove internal state/province boundaries to improve readability for peaks in Mexico and Central America.

We appreciate the reviewer's comment about making this figure clear, honest, and readable. We have removed the state/province boundaries for Mexico and Central America, as the reviewer is correct that the boundary lines are confusing in the southern parts of the study area. As to removing part of the map that do not hold peaks that were analyzed, we have added a symbol to figures 1 and 2 to clearly represent that Mt. Washington was removed from the dataset prior to analysis.

Lines 167-171: this may be placed in the Method section.

We are reluctant to make this change, for two reasons. (1) The information presented in those few lines comes from our analysis, and as such is a result of our study. (2) The temperate-tropical contrasts presented in those lines are actually highly relevant to and informative about our temperate-tropical contrasts in our treeline shift results. As such, we would greatly prefer to retain that information in its present position.

Maps: Please add the scale bar for all maps. *Done.*

Tables: Several tables presenting extensive AIC comparisons (specifically Tables 1–3 and 7–9) could be moved to the Supporting Information to improve readability and focus of the main manuscript. These tables contain numerous models with minor variations, most of which are not central to the interpretation of results and are only briefly referenced in the text. Retaining only Tables 4 and 10, which summarize the best-fitting models for treeline elevation and NDVI change, would suffice in the main text, alongside Tables 5, 6, 11, and 12 that present key model parameters and confidence intervals.

Tables 1-3 and 7-9 are now shifted to the Supporting Information, and all tables in the text and in the Supporting Information are now renumbered appropriately.

**Referee #2:**

A definition of alpine treelines (see Körner, 2012) should be provided in the introduction, as it will help to understand the overall methodological approach.

We added a simple definition of treeline for the purposes of our study to the end of the first paragraph of the introduction. Unfortunately, our institution does not have access to the Körner 2012 book, so we were unable to cite that particular source.

Moreover, insights into the remote sensing techniques employed so far to assess treeline spatial patterns would provide the context for the analyses carried out in the study.

Text added to the Next Steps and Conclusions section along these lines, citing a review paper that is particularly instructive.

The specific objectives or research questions of the work should also be clearly stated.

We have added a more specific and explicit statement of our research objectives in the final paragraph of the Introduction.

The Methods section is hard to follow due to the many steps involved in the analysis, and a diagram summarising the operations performed would greatly improve the quality of the manuscript.

We created a flow diagram that details the analysis sequentially and have added it as a new figure, Figure 4.

Many tables in the Results section should be moved to the supplementary materials as they are redundant.

We moved tables 1-3 and 7-9 from the results section to the supplementary materials. Only the model output tables remain in the main text.

The Discussion section needs to be largely improved, as a comparison with the results of other studies is far too limited, and ecological considerations are missing.

The two clear results of this study are now elaborated in greater detail in the Discussion, with at least some exploration of ecological causes of their dynamics. References to other studies added.

The comparison between Landsat imagery acquired at only two time periods, that is, the beginning of the time series (1984-1988) and a later period (2013-2017), hinders the disclosure of any temporal trajectories related to changes in vegetation cover at the treeline ecotone. Increasing the number of time periods would significantly improve the quality of the work, as this would allow the analysis of more complex temporal patterns.

We appreciate the reviewer's comment, and indeed appreciate the logic behind their suggestion. However, we explicitly have focused on long-term tendencies, which leads us to a before-and-after comparison, rather than a longitudinal tracing of changes through time. As the reviewer might imagine, before-and-after versus longitudinal data will involve very different statistics, and we found that we were able to build more sophisticated statistical models with the before-and-after comparisons. As such, in this first detailed regional overview contribution, we prefer to stick to the 4-decade comparison instead of the longitudinal tracing for each peak.

NDVI values derived from different Landsat sensors, that is, Landsat TM for the first period and Landsat ETM+/OLI for the second period, should not be compared directly unless cross-sensor spectral calibration is performed, due to the well-documented differences between Landsat sensors (see Roy et al., 2016). Moreover, normalising NDVI values based on control values, such as those extracted from stable forest cover close to each study site, would also enhance the comparability between the two study periods. See Storey et al. (2016) for an example of this.

The reviewer's concerns about spectral differences are valid. However, our analysis showed a consistent pattern across different latitudinal gradients and along different directions. The differences between Landsat 8 and Landsat 7 are around 0.024. We generate a "manual" definition of tree line. This reduced significantly the possibility of

analyzing the incorrect values or including false values. All the analyses are delimited to tree line threshold.

As all the analyses rely on absolute NDVI values, I would also suggest performing a comparison between different vegetation indices and with biophysical variables derived from Landsat imagery, such as the fractional vegetation cover. The latter could improve the robustness of the comparison between the two time periods.

We concur with the reviewer's thought about deeper exploration of different indices and data streams that can be derived from Landsat imagery. Indeed, we are planning such a more-in-depth exploration of the data streams that we have used in this study. However, this study is intended as a first exploration, and the behavior of NDVI as an approximation of photosynthetic mass is well documented, so we have elected to focus on NDVI for the purposes of this study, at least.

Many of the choices made in the methods section must be justified, as they seem arbitrary and not based on ecological considerations. For instance, it is unclear why the authors used changes in treeline elevation and NDVI as response variables in the linear models based on environmental predictors. This approach is somewhat circular, and a clear ecological rationale should be provided.

We added clear ecological rationale in the appropriate methods paragraphs, detailing why we selected our two response variables with evidence provided via additional literature citations.

The removal of transects that showed a positive relationship between NDVI values and elevation or that weren't statistically significant (Lines 116-117) greatly reduced the size of the dataset by more than a third. This approach seems somewhat arbitrary, and the type of vegetation dynamics occurring on these transects should be thoroughly investigated. As NDVI values are correlated with the amount of photosynthetic activity, treeline ecotones that have experienced marked upward shifts may have a positive relationship with elevation and thus be of interest for the study.

Though this filtering step did remove a considerable amount of data from the study, most of the original mountain peaks remained included (115 out of 120 high peaks). Because we are exploring temporal treeline movement, transects without a significant association with elevation, or treelines which do not diffuse with increasing elevation are considered anomalous in the context of our question.

Using a machine learning model, such as random forest, is advisable in this work as it would ease the analysis of the changes in both treeline elevation and NDVI, given the multidimensionality of the input dataset. Moreover, it would be beneficial for the interpretation of the importance of the predictor variables.

We appreciate the reviewer's suggestion to explore a random forest framework. However, our primary objective is to describe and quantify the relationships between environmental drivers and treeline changes, rather than to develop a predictive model. Linear mixed effects and spatial mixed models provide interpretable coefficients and confidence intervals. Moreover, our mixed models explicitly incorporate spatial covariance structures to address spatial autocorrelation, whereas random forest does not inherently adjust for spatial dependence. We have also conducted thorough diagnostics (e.g., residual

checks, VIF assessments, and Moran's I tests) and found that our models satisfy all assumptions and explain a substantial part of the variance in both response variables. Introducing a random forest analysis would likely confirm driver importance but add little to our mechanistic understanding of driver effects. Thus, we respectfully decline to add a random forest analysis, as our current modeling approach best serves the goal of elucidating how spatial variables drive treeline dynamics.

- Line 8. Please compare the number of analysed peaks with that reported at lines 51 and 78. We changed 119 to 115 in line 51, as that was an error that we thank the reviewer for observing. However, in line 78, we reference the number of peaks remaining at that point in the data preparation process. We explain in the next section how the number of peaks analyzed was reduced from 120 to 115. We hope that the addition of the flowchart additionally clarifies this point.
- Lines 13 15. Please modify the end of the sentence, as the statement seems speculative. *Done.*
- Line 51. Please specify what type of treeline shifts have been investigated. *Done.*
- Line 52. Please clarify what "vegetation index profiles" mean in this context.

  We were referring to how NDVI trends along the transects radiating out from each peak. We have now reworded to provide greater clarity.
- Lines 53 55. This sentence needs to be improved as the limitations associated with the spatial resolution of remote sensing data and the benefits of vegetation indices appear to be mixed.

We are a bit at a loss with this comment, as the paragraph is not about the limitations associated with spatial resolution of the remote sensing data, but rather about how the RS data are complementary to on-the-ground studies that provide much greater detail but that lack the generality and broad-scale scope that the RS data allow. In sum, we believe that this paragraph "works" given its purpose.

Line 61. It is unclear how the authors measured the vegetation mass at the treelines. NDVI provides adimensional values bounded between -1 and 1 that are associated with the photosynthetic activity of vegetation. The index doesn't provide any specific information on biomass.

We agree with the reviewer that NDVI does not provide direct biomass measurements. However, in our treeline delimitation, the highest NDVI values serve as a proxy for tree presence. We manually delineated these limits based on our observations in the field. Additionally, the highest NDVI values correspond to the forest line.

Lines 68 – 70. Please replace the details with the projection EPSG code. *Done.*

- Line 70. Please motivate the choice of the sampling distance along the transects. *Text added to clarify.*
- Lines 72 73. Please clarify how the presence of anthropogenic activity was assessed. *Text added to clarify.*
- Lines 73 75. Does this relate to latitudinal treelines? Please provide a clear explanation. We added a clause clarifying that we are referring to the removal of latitudinal treelines from the study.
- Lines 81 83. Please provide more details about the Landsat-derived NDVI dataset (Landsat Collection, data provider, preprocessing level, etc.). It is unclear what is the Landsat annual collection.

We added additional details on the Landsat-derived NDVI dataset

Lines 83 – 84. It is unclear whether the authors used a pre-processed dataset containing annual NDVI values (as described in the previous sentence) or if it was computed from yearly Landsat reflectance image composites. Please provide more details related to Landsat data (Collection, preprocessing level, cloud masking, etc.).

Thank you for pointing this out. We used the pre-processed LANDSAT\_LT5\_L1T\_ANNUAL\_NDVI collection in Google Earth Engine, which provides annual NDVI composites derived from Level-1 terrain-corrected Landsat 5 reflectance images (including cloud masking and quality assurance). Unfortunately, this collection is now deprecated in GEE. To ensure full transparency, our scripts for reproducing the NDVI computation from the original Landsat reflectance data are publicly available [link to scripts]. Detailed information about the original dataset can be found here:

https://developers.google.com/earth-engine/datasets/catalog/LANDSAT\_LT5\_L1T\_A NNUAL\_NDVI

Calibration procedures and validation methods for this collection are described in: <a href="https://www.sciencedirect.com/science/article/abs/pii/S0034425709000169">https://www.sciencedirect.com/science/article/abs/pii/S0034425709000169</a>

We included these details and links in the revised Methods section to clarify our workflow.

Lines 87 - 90. This approach describes the manual detection of the timberline instead of that of the treeline ecotone. Please see the relevant definitions in Körner (2012). A quantitative approach is needed here, such as the percentage of tree canopy cover within a single Landsat pixel.

We concur with the reviewer that an objective approach would be optimal for this step. However, as we are still in the process of developing that approach (it will eventually be a deep learning algorithm, which we have under development), we have relied on visual evaluation for this first report. We believe that our protocol is described in good detail as it currently stands.

Lines 91 - 94. The approach is somewhat opaque and needs clarification. Why did the authors end the transects after up to 20 stations? How did the authors assess the presence of anthropogenic effects?

Text added to clarify, but description is pretty detailed as it is.

Lines 96-97. It is unclear what the difference is between NA and missing values. How were raster values extracted from the sampling stations?

Clarified in text.

Lines 97 – 99. Averaging NDVI values over the two study periods doesn't reduce the influence of outliers, such as those caused by unmasked cloud cover, as this statistic is not robust to the influence of extreme values. What do the authors mean by "random effects"?

Reworded to reflect the reviewer's comment.

Lines 100 – 101. Vegetation mass doesn't appear appropriate. See the previous comment for line 61.

We replaced "vegetation mass" with "NDVI".

Lines 101 – 102. It is unclear what a hierarchical nested list of lists is. I suggest omitting this information.

This information is a crucial aspect of our data preparation for the analysis. However, we added a panel within the newly created flowchart to show how the data is structured to provide further clarification.

Lines 102 – 104. Please specify which criteria were used for entirely removing some transects, as section 2.2 doesn't provide any.

This was a typographical error; we intended to refer back to section 2.1. We corrected this error.

Lines 108 – 109. Geographic noise doesn't seem to be the appropriate term here. Please rephrase the sentence.

Done.

Lines 110 - 112. Which is the response variable between NDVI and elevation? Please indicate it. Please rephrase the sentence, as "the best regression equation" and "the best approximation to the relationship between these variables" seem to have the same meaning. It is unclear which is the "final response variable" from Figure 2; please indicate it clearly in the text.

We further clarified these sentences and explicitly stated the response variable. We also move the reference to figure 2 to earlier in the sentence to avoid confusion.

Figure 2. The number of points with a negative slope in the map seems much lower than the number indicated in the text (694). Please include a legend for the values associated with the size of the circles, together with a scale bar.

We explain in the figure 2 caption that the 1-8 transects are averaged for each peak, hence the reduced number of points compared with the number of transects reported in the text. We added scale bars to both maps and a legend item that includes values associated with point size.

Lines 113 – 115. It is unclear why the authors used the AIC as a metric for comparing the models. The coefficient of determination, root mean squared error, and the Bayesian information criterion are all suitable performance measures.

We appreciate the reviewer's suggestion to consider alternative metrics, but we selected AIC because our primary goal is to identify the most parsimonious explanatory model rather than optimize predictive accuracy. AIC directly balances goodness-of-fit and model complexity for likelihood-based inference, whereas RMSE is most appropriate when assessing out-of-sample predictive performance rather than hypothesis testing. Although BIC imposes a somewhat harsher penalty for additional parameters, it is derived under a Bayesian framework, which we do not adopt elsewhere in our analysis; using BIC here would introduce an inconsistent inferential basis. Consequently, AIC remains the most suitable criterion for comparing competing explanatory models in our study.

- Lines 116 117. What level of statistical significance was considered by the authors? We added a sentence stating that  $\alpha$  = 0.05 was out threshold for statistical significance.
- Lines 117 119. Please specify what criteria were used to eliminate the transects. We clarified that we are referring to the criteria outlined in the previous sentences.
- Lines 121 122. This sentence should be moved to the beginning of the paragraph (Line 110).

These data refinement steps are steps along the way to calculating the response variables near line 110, which was clarified as part of a separate comment. As such, we feel that lines 121-122 provide crucial insight as this is where we explain how we calculated our response variables. Thus, we believe that it is much better to leave the sentence at lines 121 and 122, as it is.

Lines 122 – 123. Please rephrase this sentence as it is unclear. Do the authors mean that they predicted elevation based on NDVI values obtained for the most recent period? *Done*.

Lines 123 – 127. Estimating a biophysical variable, such as canopy cover, from Landsat data would be a more robust approach. Changes in NDVI values are related to many external factors, such as differences in the spectral response between Landsat sensors. A validation of the methodology using high-resolution data, such as historical aerial orthophotos, is also advisable.

We agree, and we appreciate the comment from the reviewer. We plan such in-depth analyses in the future, but for the moment we prefer the relative simplicity of NDVI as a proxy for photosynthetic mass, also a biophysical variable. We have included mention of this complication, as well as a relevant reference, in the Discussion.

Lines 143 – 145. Please rephrase the sentences. It doesn't seem that the first approach is suitable for assessing spatial autocorrelation.

We thank the reviewer for identifying this oversight. The sentence has been broadened so that it is inclusive of all three types of statistical models employed in the study.

Lines 144 - 145. It is unclear why the models in the Results section are 18 instead of 16, as indicated in the sentence.

We corrected all instances of 16 to 18, as 16 models is an outdated count.

Lines 147 – 148. Please specify the R package used for this analysis. *Done.*

Lines 149 – 145. Please remove this paragraph as none of the results are included in the manuscript.

We removed the paragraph.

Lines 163 – 164. Please provide a reference for this approach. *Done.*

Lines 173 – 175. An assessment of the results based on high-resolution imagery (historical and current) and performed on a subset of the peaks would greatly improve the robustness of the methodology.

We agree with the reviewer that high-resolution imagery is a great method for assessing treeline dynamics. However, integrating rephotographic methods would entail a separate workflow and validation framework, which is outside the scope of this manuscript. We therefore prefer to maintain methodological consistency by focusing on NDVI-based proxies in the present analysis and reserve detailed rephotographic assessments for a follow-up study.

Lines 176 - 179. The coefficient of determination should also be provided to evaluate the performance of the models in absolute terms.

We added the coefficient of determination.

Lines 179 - 182. The results in Tables 1 to 3 should be moved to the Supplementary Material, and only the results relative to the best models should be reported in the Results section.

Reporting the results from supplementary tables functions to describe the predictor variables included in Table 4 (Table one in the revised manuscript), which compares the best linear mixed model and the best spatial mixed model. We therefore find it necessary to retain these results for clarity.

Lines 188 – 189. Please rephrase the sentence. The residual variance is not reported in Table 5 as stated in the sentence. The caption in Table 5 needs to be checked, as none of the numbers are in bold text.

We added the residual variance to table 5. None of the numbers are in bold text because none of the predictor variables were significant in this model; we updated the caption to state this explicitly for clarity.

Line 199. Please move Tables 7, 8, and 9 to the Supplementary Material.

Tables 7, 8, and 9 have been moved to the supplementary materials

Lines 223 – 225. Please elaborate more on this. What are the effective methods the authors are referring to?

Clarified.

Lines 227 – 231. This is the main finding of the work and should be discussed in more detail by providing the ecological context and a summary of the findings of previous studies.

As detailed above, we have added some text to speak to this comment, and a previous summary comment that was similar. We strive not to over-discuss, so the additions are brief, but we hope that they will speak to what the reviewer is after.

Lines 237 – 238. This sentence is a little speculative. Please rephrase or remove it. Sentence removed.

Lines 240 – 242. Please note that Landsat imagery covers more than five decades when considering data acquired by the MSS sensor onboard Landsat 1-5 satellites, i.e., since 1972.

We appreciate that Landsat imagery actually crosses more than 50 years, but our analyses only span three decades. We have reworded for clarity.

**Remarks from preceding review file validation:**

a) Author affiliations 1 and 3: please add the city and country. b) Section "Author contributions": please use initials for the authors' names. c) Section "Code and data availability": please start the URL in a new line so that the complete link is visible. d) Figure 5: please add the © icon to "Google Earth".

We addressed all remarks aside from adding the city to affiliation 3. Laboratorio Nacional CONAHCYT de Biología del Cambio Climático is a group of institutions in Mexico in different cities, thus, it does not have one representative city to add to the affiliation.